# Forgiveness and Flourishing: The Mediating and Moderating Role of Self-Compassion

**DOI:** 10.3390/ijerph20010666

**Published:** 2022-12-30

**Authors:** Justyna Mróz

**Affiliations:** Department of Psychology, Jan Kochanowski University of Kielce, 25-369 Kielce, Poland; jmroz@ujk.edu.pl

**Keywords:** forgiveness, self-compassion, psychological well-being, well-being, flourishing

## Abstract

(1) Background: This study investigated the relationships between forgiveness, self-compassion, and flourishing, and examined the mediating and moderating role of self-compassion (self-warmth and self-cold) in the relationship between forgiveness and flourishing. (2) Methods: A sample of 300 Polish participants aged 18–57 (*M* = 23.53 years, *SD* = 5.82) completed the Heartland Forgiveness Scale, the Self-Compassion Scale, and the Flourishing Scale; we used Spearman’s rho correlations to assess the associations between the main analyzed variables and used PROCESS software to calculate moderation and mediation. (3) Results: The obtained data showed that forgiveness and self-compassion were positively related to flourishing. Self-warmth (positive dimension of self-compassion) mediated and moderated the link between forgiveness and flourishing. In contrast, self-coldness (negative dimension of self-compassion) did not mediate or moderate the association between forgiveness and flourishing. (4) Conclusions: The results suggest that positive resources relate to and support one another. Compassionate self-responding is associated with positive resources; in contrast, uncompassionate self-responding is not significant for positive variables.

## 1. Introduction

Achieving well-being is one of the most important human goals. Research into mental health viewed in positive terms leads to better understanding of the indications that build well-being. Flourishing is a positive term, which means to function in a way that is conducive to growth, resilience, and goodness [1]. Flourishing links to both hedonist and eudaemonist components of well-being, and includes concepts such as purpose in life, relationships, self-esteem, feelings of competence, and optimism [2].

Understanding the factors that support flourishing can help design prevention models that reduce negative health symptoms among different groups. Drawing on the existing theory and research in positive psychology, in this study we proposed forgiveness and self-compassion as independent variables and flourishing as an outcome variable.

### 1.1. Forgiveness and Psychological Well-Being

In this paper, forgiveness is conceptualized as a general propensity to forgive, regardless of time, relationships, and situations [3,4]. Forgiveness is also a personality trait which involves prosocial emotions such as love, sympathy, compassion, and/or a reduction in negative emotions such as anger or hostility [4]. Additionally, forgiveness is associated with positive motivation such as benevolence towards a wrongdoer and/or reduced negative motivations, such as avoidance or revenge [5].

One of the theoretical approaches explaining the link between forgiveness and flourishing is the stress-and-coping model of forgiveness [6]. This model is based on the transactional theory of stress developed by Lazarus and Folkman [7]. The model of stress-and-coping of forgiveness recommends that coping through forgiveness is one of the more efficient forms stress reduction and positive adaptations to harm [8,9].

Another model referring to the relationship between forgiveness and well-being is the scaffolding self and social systems model of forgiveness and well-being (4S model) [10]. Forgiveness leads to subjective well-being through relationship harmony, relationship mastery, adaptive identity development, and self-acceptance/self-worth. This model suggests that forgiveness leads to an increase in positive perceptions of self and others. This is consistent with the broaden-and-build-theory [11], which suggests that, for example, the experiences of relief and other positive feelings which come after forgiveness builds other positive resources, leading to psychological well-being.

Empirical work on forgiveness and psychological well-being has showed positive associations between them. Toussaint and Fridman [9] used a number of different tools to measure both forgiveness (Heartland Forgiveness Scale, HFS, and Transgression-Related Interpersonal Motivations Scale, TRIM) and well-being (Satisfaction with Life Scale, SWLS, The Fordyce Happiness Scale, and The Bradburn Affect Balance Scale) in psychotherapy outpatients. Their results showed that, regardless of the tool used, the relationship between forgiveness and well-being was significant and positive. Additionally, a study among Ukrainian war refugees indicated a positive correlation between forgiveness (Decision to Forgive Scale, DTFS), mental well-being (The World Health Organisation Five Well-Being Index, WHO-5), and spiritual well-being [12]. Other studies in more general groups have also confirmed this link [13,14].

### 1.2. Self-Compassion as a Mediator or a Moderator

The relationship between forgiveness and well-being is well established in the literature [9,13]. Continuing to seek mechanisms and mediators to explain this relationship will further understanding. Previous studies have pointed to the mediating role of the affect and beliefs [9] and feeling connected to others [15], between forgiveness and well-being.

Referring to the scaffolding self and social system model of forgiveness [10] and the stress-and-coping model of forgiveness [6], self-compassion can be a potential mediator in a casual pathway between forgiveness and psychological well-being.

Self-compassion is a dynamic system that supports coping with difficult events. It comprises three components: (1) the first of them refers to an emotional response to suffering (with kindness or judgment); (2) the next refers to the cognitive assessment of one’s own difficult situation as an experience of common humanity or as isolation; (3) the third component is about perceiving suffering (with mindfulness or over-identification) [16]. The positive components of self-compassion—self-kindness, common humanity, and mindfulness—are described as self-warmth. The negative components of self-compassion—self-judgment, isolation, and over-identification—are referred to as self-coldness [17]. Self-warmth is consistent with positive psychology, and it supports the protective role of self-compassion. The positive subscales of self-compassion related to more positive variables are gratitude, hope, and self-esteem.

On the other hand, uncompassionate self-responding—self-coldness (self-judgment, isolation, and over-identification)—is associated with symptoms of psychopathology such as anxiety disorders, depression, and other mental health problems [17].

Previous studies have found that higher levels of self-compassion are linked to other positive psychological constructs, such as life satisfaction [18], gratitude [19], and resilience [20]. On the other hand, lower levels of self-compassion are related to depression [17,21,22], anxiety [23], and PTSD symptoms [24].

Recent studies have shown that a self-compassionate orientation might help cope with negative situations, emphasizing its buffering role [17]. In the present study, self-compassion is presented as a variable supporting the relationship between forgiveness and well-being in two possible roles, as a moderator and as a mediator. Self-compassion as a moderator might be an enhancer of well-being, in that it highlights the positive implications of forgiveness. On the other hand, forgiveness as a mediator might increase flourishing by enhancing self-compassion. Self-compassion can be conducive to both the hedonic and the eudemonic aspects of flourishing. The former shows that self-compassion can increase positive emotions and subjective well-being [16], whereas the latter focuses on the supporting role of self-compassion in utilizing adaptive mechanisms in difficult situations [25].

The mediation model hypothesizes that the components of self-compassion mediate the relationship between forgiveness and flourishing. Many studies have focused on the mediating role of self-compassion between negative variables [17,26] or buffering positive outcomes in negative situations [27,28].

Despite the lack of studies where self-compassion mediates the link between forgiveness and well-being, previous studies have reported significant mediation relationships with self-compassion as the mediator of positive resources and well-being as an outcome variable (mindfulness–psychological well-being) [29]. The theory combining forgiveness, self-compassion, and flourishing proposed by Hobfoll, called the resource caravan passageways, indicates that resources travel in packs or caravans and support each other [30]. Combined positive resources lead to positive mental health, and support coping with difficult events [31].

Several studies have examined the moderating role of self-compassion. These studies have found that self-compassion buffers [32] and supports [33] positive human functioning. Self-compassion moderates the link between dietary restraint and emotion-focused impulsivity. This link is weaker for individuals with higher levels of self-compassion [32], pointing to the buffering role of self-compassion. Chen [33], examining the relationship between PsyCap and life satisfaction, found a supporting role in self-compassion among students.

### 1.3. The Present Study

Much of the early research focused on negative links between forgiveness and self-compassion and negative outcomes, such as anxiety, depression, and PTSD [34,35]. Little research has been conducted to examine the possible underlying impact of these variables on the positive side of life [36]. According to the tenets of positive psychology, fostering positive aspects of functioning (e.g., psychological well-being) is just as important as preventing negative consequences, such as depression, anxiety disorders, etc.

Referring to the scaffolding self and social systems model of forgiveness and well-being [10] and the stress-and-coping model of forgiveness [6], both forgiveness and self-compassion could support flourishing. Forgiveness and self-compassion could also weaken the aftermath of negative events by emotional and cognitive reframing, reducing negative feelings, thoughts, and behaviors. Through forgiveness, individuals who are victims can reformulate negative emotions, thinking, and motivation from negative to neutral or positive [4]. On the other hand, individuals with high self-compassion do not replace negative emotions and thoughts with positive ones; they accept negative events and give new meaning to them [16,25]. This is consistent with both hedonist and eudaemonist theories of psychological well-being.

Based on the reviewed theory and research, we examined whether two dimensions of self-compassion mediated and moderated the association between forgiveness and flourishing. The hypotheses were as follows: (1) forgiveness would inter-relate to increase flourishing through increased self-warmth and decreased self-coldness; (2) the link between forgiveness and flourishing would be stronger with higher self-warmth and lower self-coldness.

## 2. Materials and Methods

### 2.1. Power Analysis

To determine the minimum sample size for the current study, the G*Power version 3.1. Program [37] was used. The sample size required for multiple regression analyses with three independent variables for detecting a medium effect (f^2^ = 0.03) with a power of 0.80 and 0.05 level of significance was *N* = 204 or more. To avoid Type II errors, the bootstrapped samples in the PROCESS macro were set to 5000 at 95% bias-corrected confidence intervals, which was statistically adequate for the number of respondents.

### 2.2. Participants

We used a sample of 300 adult participants from Poland. Female participants accounted for 83.3% (n = 250) of the sample. The subjects’ age ranged from 18 to 57 years, with a mean of 23.53 (*SD* = 5.82). Regarding the level of education, 1% of the sample had completed primary education, 1% had completed vocational education, 41.4% had completed secondary education, 22.7% had a university degree, and 34.2% had graduated from college. The respondents were requested to participate in the study voluntarily—no remuneration was offered to them. Data were collected between October 2021 and February 2022 using an online questionnaire distributed via social networking sites. All respondents provided informed consent online. The responses were anonymous, and the confidentiality of information was assured. Participants were informed about their right to terminate the survey at any time if they wanted.

### 2.3. Methods

#### 2.3.1. Forgiveness

Disposition to forgive was measured using the Heartland Forgiveness Scale [4]. The HFS is a multidimensional tool assessing the dispositional forgiveness of self, others, and situations beyond one’s control. Participants rate their responses to 18 items on a 7-point scale. Higher scores on each scale reflect higher levels of forgivingness. The total HFS score indicates how forgiving a person tends to be. In this study, we only used the total score. The Cronbach’s alpha (internal consistency) for total HFS was 0.85 in this study.

#### 2.3.2. Self-Compassion

Self-compassion is typically assessed using the Self-Compassion Scale (SCS) [38]. The original SCS has 26 items measuring six components of self-compassion in two dimensions. The first dimension of self-warmth includes self-kindness, common humanity, and mindfulness. The second dimension of self-coldness includes self-judgement, isolation, and over-identification. Items are rated on a 5-point scale ranging from 1 (almost never) to 5 (almost always). Test–retest reliability was established as good for the overall scale (r = 0.87, *p* < 0.01, Cronbach’s alpha = 0.93), as well as the subscales (Cronbach’s alpha = 0.80–0.89).

#### 2.3.3. Flourishing

Flourishing was measured with a brief 8-item Flourishing Scale (FS). The range of scores is from 8 to 56, where higher scores mean a higher level of psychological well-being [2]. The FS has demonstrated good validity in different cultures. The Cronbach’s alpha coefficient was 0.91 in this study.

### 2.4. Data Analysis

Before the beginning of the main analysis, data were screened for potential errors in the expected range of values and for any indicators of careless answers. We used the Mahalanobis distance to evaluate the outliers [39]. All results fulfilled the criteria. All observations (*N* = 300) were included in the main statistical analysis. The studies were completed online, which avoided deficiencies. Incomplete answers were not included in the data file. We used Spearman’s rho correlations to assess the associations between the main analyzed variables: forgiveness, self-compassion, and flourishing. We used IBM SPSS software (version 26, PS IMAGO PRO 6.0, Predictive Solutions) and employed regression-based analysis to directly test the proposed moderating model and the mediating model using PROCESS software [40]. Self-compassion was both a moderator and a mediator. Model 1 (moderating analysis) and 4 (mediating analysis) were estimated using PROCESS with 5000 bootstrap samples and 95% bias-corrected bootstrap intervals for all indirect effects. For all data, the hypothesis of a normal distribution of the measurement results was tested using the Kolmogorov–Smirnov test; the results showed that the data were not normally distributed.

## 3. Results

### 3.1. Preliminary Analyses

The results of the correlational calculations demonstrated that most of them were statistically significant (Table 1). We used Spearman’s rho to calculate correlations. Forgiveness was positively and significantly correlated with flourishing and self-compassion (total score, self-kindness, common humanity, mindfulness, and self-warmth), and inversely correlated with four subscales of self-compassion: self-judgment, isolation, over-identification, and self-coldness. Flourishing was positively and significantly correlated with three subscales of self-compassion—self-kindness, common humanity, and mindfulness—and negatively and significantly correlated with self-judgement, isolation, and over-identification.

### 3.2. Mediational Analyses

To examine whether the two dimensions of self-compassion mediated the association between forgiveness and flourishing with age and gender as covariants, we used a multiple mediation model (Model 4 in PROCESS). All outcomes were standardized. Forgiveness was linked to all mediators—it showed a positive correlation with self-warmth (β = 0.64, *p* < 0.001), and an inverse correlation with self-coldness (β = −0.66, *p* < 0.001) (Figure 1). Only one mediator was significantly related to flourishing—self-warmth (β = 0.41, *p* < 0.001). The indirect effect (IE) of forgiveness on flourishing via self-warmth and self-coldness was found to be significant, because the 95% confidence interval did not include zero (β = 0.26 CI95% [0.157, 0.363]). However, only self-warmth was a significant mediator (IE, β = 0.26; CI95% [0.189, 0.344]). The indirect effect accounted for 50.49%.

This model was a good fit with the data (∆*R*^2^ = 0.36, *F*(294, 5) = 32.861, *p* < 0.001).

### 3.3. Moderating Analysis

To test whether self-compassion (two dimensions: self-warmth and self-coldness) moderated the relationship between forgiveness and flourishing, we used a one-model moderation analysis in PROCESS. Age and gender were using as covariants. All outcomes were standardized.

The analysis revealed a significant interaction of forgiveness and self-warmth with flourishing (∆*R*^2^ = 0.37, *F*(294, 5) = 34.192, *p* < 0.001). Forgiveness was positively related to flourishing at a low level of self-warmth (β = 0.19, CI95% [0.108, 0.282]), an average level of self-warmth (β = 0.13, CI95% [0.064, 0.211]), and a high level of self-warmth (β = 0.09, CI95% [0.001, 0.180]). Higher self-warmth was linked with a stronger positive relationship between forgiveness and flourishing when compared with low levels of self-warmth (Figure 2). In contrast, forgiveness did not predict a change in flourishing for self-coldness (β = 0.002; CI95% [−0.002, 0.007]).

## 4. Discussion

In this study, mediation and moderation models have been proposed, in which self-compassion mediates and moderates the relationship between forgiveness and flourishing. This study investigated the differences in mediation and moderation between two dimensions of self-compassion: self-warmth and self-coldness.

The results obtained here partially support the hypothesis that forgiveness will inter-relate with increased flourishing through increased self-warmth and decreased self-coldness. Our data showed a stronger tendency to forgive, inter-related with higher levels of flourishing via higher levels of self-warmth. Forgiveness, especially self-forgiveness, is understood as a manifestation of a positive attitude towards oneself even when one has been disappointed in oneself and others, which is consistent with self-warmth. Therefore, forgiveness with self-warmth had a strong effect on well-being.

Our findings correspond with previous research showing that the positive relationship between forgiveness and well-being is mediated by other positive variables [15]. For example, Bono et al. [15] found closeness to be a mediator between forgiveness and well-being in psychology students. Forgiveness was measured as a negative motivation (avoidance and revenge) and positive motivation (benevolence), and well-being was measured as a subjective assessment of life satisfaction, which is consistent with the hedonistic approach. Higher benevolence and higher closeness resulted in evaluating life as satisfying. On the other hand, self-warmth was a mediating association between mindfulness and personal recovery [41].

Interestingly, in the current study, self-coldness was not a significant mediator between forgiveness and well-being. Previous research has indicated the significant indirect effects of self-coldness on mental health (e.g., depression) [17]. Brophy et al. [17] also found that self-coldness mediated the association between attachment and depression, and that it had a stronger effect than self-warmth. Additionally, Lu et al. [42] found that self-coldness was a stronger mediator in the relationship between stigma and two variables—depressive symptoms and demoralization—in hemodialysis patients. Possibly, self-coldness as a negative dimension of self-compassion is a stronger predictor of psychopathology than self-warmth. In the present study, positive variables were used; thus, a stronger effect is shown by the positive dimension of self-compassion. Similar results were found by Mak et al. [41], where self-warmth, but not self-coldness, was a mediator between two positive variables—mindfulness and personal recovery. Self-warmth as an emotional regulation strategy [38] can reduce negative emotions, such as negative behavior towards a wrongdoer of an offence, which can lead to increased well-being. On the other hand, forgiveness is the letting go of negative emotions towards a wrongdoer and showing benevolence oneself [43]. Thus, there is a strengthening of self-warmth and a psychological well-being.

Our second hypothesis about self-warmth and self-coldness functioning as moderators between forgiveness and flourishing was partially corroborated. Self-warmth, but not self-coldness, was a moderator. Regardless of the level of self-warmth (low, medium, or high), forgiveness was positively associated with flourishing. In other words, exhibiting warmth to oneself helps people increase flourishing by forgiveness. The moderation outcomes regarding forgiveness and well-being are consistent with previous studies revealing associations between forgiveness and well-being, including the hedonistic approach and the eudaimonia theory [13,14]. The result is also supported by the suggestion that the positive dimension of self-compassion can strengthen the link between forgiveness and well-being. This is consistent with compassion-focused therapy, which assumes that capacities of warmth and care towards oneself enhance well-being [44]. Additionally, Gilbert [44] proposed that self-compassion affects well-being by activating the social-safeness neurological system and deactivating the threat-defense system. Furthermore, self-compassion training decreases sympathetic nervous system reactivity and enhances adaptive parasympathetic activity [45].

In contrast, the negative dimension of self-compassion did not moderate the association between forgiveness and flourishing. Self-coldness may be a more effective moderator in the context of negative indicators of mental health. Our results are supported by previous studies showing that self-warmth and self-coldness have different interaction mechanisms [17]. Compassionate self-responding—self-warmth, including self-kindness, common humanity, and mindfulness—fosters the link between positive resources. On the other hand, uncompassionate self-responding—self-coldness (self-judgment, isolation, and over-identification)—is associated with symptoms of psychopathology [46,47].

This argument is supported by the Conservation of Resources theory and the resources caravan theory [30,31]. The resources support each other; they travel in caravans, not in isolation; and they are associated with other resources. The loss of some resources causes the loss of the next resources. Similarly, strong, positive resources foster growth in other positive resources.

Previous research has focused particularly on the mediating role of self-compassion for negative variables, such as depressive symptoms [17,21,22,42], suicidal risk [48], anxiety [49,50], and personality disorders [51,52]. Our results support the few previous studies focusing on positive variables [41], showing that self-compassion plays an important role in mental health and well-being. This explains that the application of methods such as compassion-focused therapy [44] or mindful self-compassion [53] can not only be employed in the treatment of disorders, but also as a method of prevention or reinforcement of positive aspects of mental health.

The inter-relationships between forgiveness, self-compassion, and flourishing can be interpreted in light of the scaffolding self and social systems model of forgiveness and well-being (4S model) [10]. According to this model, forgiveness of oneself and others should entail stronger positive attitudes to oneself and others, such as self-acceptance and self-esteem, and leads to enhanced well-being. Tendency to forgive fosters kindness towards oneself by perceiving oneself as a moral person.

There are limitations to the present research that warrant attention. Firstly, only self-reporting tools were used, and all tools measured trends, not the present state. Future studies should include observer-rated variables or tools which measure variables as states (and not only traits). Secondly, due to the cross-sectional design, no causal inference can be made. Longitudinal designs or experiments in future studies should be used to confirm this causality. Thirdly, this study was based on data from a small sample. Future investigations could utilize a more heterogeneous group in terms of age, culture, clinical problems, etc. Next, this study only concerned positive aspects of mental health. This may have limited determining the mediating role of self-compassion. The design of future research should consist of both aspects of mental health—positive well-being and negative well-being, such as depression, anxiety, or feelings of stress. This study is one of the first to focus on the mediating role of self-compassion between forgiveness and flourishing; therefore, further research on this issue is necessary to better understand this mechanism.

Finally, the current investigation could be replicated with other variables controlled which may also mediate or moderate the relationship between forgiveness and flourishing/well-being.

## 5. Conclusions

The relationship between forgiveness and psychological well-being is well-documented. However, studies are exploring the underlying mechanisms of this relationship. The presented outcomes show that self-warmth, not self-coldness, is a mechanism (variable) which could explain the inter-relation between forgiveness and flourishing. Additionally, this conclusion is important in the context of previous studies which concern the mediating and moderating role of self-compassion, such as earlier studies including negative symptoms in mental health [27]. Our findings suggest that self-warmth is a more effective mediator and moderator between positive variables. In contrast, self-coldness has a stronger effect than self-warmth in negative variables, which resulted from previous data [17]. These results highlight differences between the dimensions of self-compassion. Self-warmth as a compassionate self-response supports the development of other resources, which buffer mental health.

Additionally, in practice, when designing positive interventions, the supporting role of self-warmth can be used to strengthen other positive resources such as forgiveness and flourishing.

## Figures and Tables

**Figure 1 ijerph-20-00666-f001:**
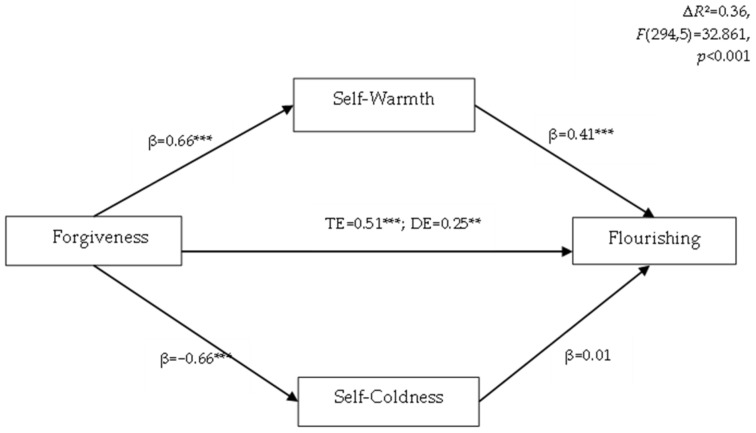
Parallel mediation model of forgiveness on flourishing using self-warmth and self-coldness as mediators. Standardized coefficients are presented. ** *p* < 0.01, *** *p* < 0.001; TE—total effect; DE—direction effect.

**Figure 2 ijerph-20-00666-f002:**
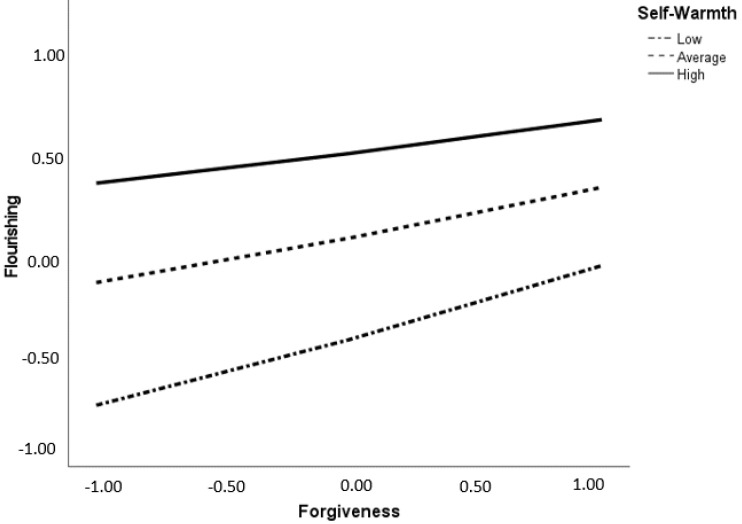
A visual representation of the moderation effect of forgiveness on flourishing at low, average, and high levels of self-warmth. The graph shows standardized data.

**Table 1 ijerph-20-00666-t001:** The bivariate correlations (Spearman’s rho) between forgiveness, flourishing, and self-compassion.

		1	2	3	4	5	6	7	8	9	10	11
1.	Forgiveness	-										
2.	Self-kindness	0.628 **	-									
3.	Common humanity	0.479 **	0.605 **	-								
4.	Mindfulness	0.555 **	0.711 **	0.534 **	-							
5.	Self-warmth	0.653 **	0.927 **	0.804 **	0.836 **	-						
6.	Self-judgment	−0.593 **	−0.594 **	−0.340 **	−0.405 **	−0.545 **	-					
7.	Isolation	−0.566 **	−0.434 **	−0.355 **	−0.408 **	−0.465 **	0.577 **	-				
8.	Over identification	−0.586 **	−0.513 **	−0.406 **	−0.458 **	−0.544 **	0.725 **	0.671 **	-			
9.	Self-coldness	−0.661 **	−0.589 **	−0.413 **	−0.477 **	−0.589 **	0.892 **	0.839 **	0.893 **	-		
10.	Self-compassion	0.736 **	0.850 **	0.690 **	0.741 **	0.894 **	−0.794 **	−727 **	−0.799 **	−0.881 **	-	
11.	Flourishing	0.521 **	0.558 **	0.476 **	0.450 **	0.588 **	−0.328 **	−0.447 **	−0.358 **	−0.431 **	0.578 **	-

** *p* < 0.01.

## Data Availability

The dataset presented in this study is available on reasonable request from the corresponding author.

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
