# Peer review of "Forgiveness and Flourishing: The Mediating and Moderating Role of Self-Compassion"

_ijerph, 2022, doi:10.3390/ijerph20010666_

Round 1
Reviewer 1 Report (Previous Reviewer 2)
The authors made a lot work to improve this work. All the comments are addressed. I have no other suggestion.
Author Response
Thank you very much for your comment and the work you put into reviewing my article.
Reviewer 2 Report (Previous Reviewer 1)
Dear author,
Thanks for revising you manuscript and addressing all comments!
Author Response
Thank you for your comment and the work you put into reviewing my article.
This manuscript is a resubmission of an earlier submission. The following is a list of the peer review reports and author responses from that submission.
Round 1
Reviewer 1 Report
Thank you for the opportunity to review this interesting and timely manuscript entitled “Forgiveness and Flourishing. The Mediating and Moderating Role of Self-Compassion” submitted for publication in International Journal of Environmental Research and Public Health. I consider the manuscript as a valuable and underlying work. There are some parts that need more clarification in different sections as follows. I believe the manuscript can be accepted after minor revision (corrections to minor methodological errors and text editing).
Abstract:
· M and SD are missing from the abstract.
“A sample of 300 Polish participants aged 18-57 (M=years, SD=..)…”
1. Introduction:
· There are couple of abbreviations that are not defined, such as HFS, TRIM, SWLS, etc.
2. Materials and Methods
· There is no information provided about the procedure for survey delivery and data collection.
Participants:
· The target sample was not defined well. Author just mentioned 300 participants from Poland, but it is not clear if any specific characteristics of these participants were involved in selection.
3.2. Mediational Analyses
· It seems there is a typo in the indirect effect. B=26.
“The indirect effect of forgiveness on flourishing via self-warmth and self-coldness was found to be significant, as the 95% confidence interval did not include zero (B=26. CI95%[.157, .363]).”
· What does IE stand for in the following sentence?
“However only self-warmth was significant mediator (IE, B=.26; CI95%[.189, .344]).”
What does IE stand for in the following sentence?
“However only self-warmth was significant mediator (IE, B=.26; CI95%[.189, .344]).” I believe the manuscript can be accepted after minor revision (corrections to minor methodological errors and text editing).
Author Response
Reviewer1
Thank you very much for your review. We appreciate your suggestions and comments, which will undoubtedly improve the quality of our manuscript.
1.Abstract: M and SD are missing from the abstract.“A sample of 300 Polish participants aged 18-57 (M=years, SD=..)…”
Answer: I have completed the missing data
- Introduction: There are couple of abbreviations that are not defined, such as HFS, TRIM, SWLS, etc.
Answer: The abbreviations are explained
- Materials and Methods: There is no information provided about the procedure for survey delivery and data collection
Participants:
The target sample was not defined well. Author just mentioned 300 participants from Poland, but it is not clear if any specific characteristics of these participants were involved in selection.
Answer: I have completed missing information.
- Mediational Analyses
- It seems there is a typo in the indirect effect. B=26. “The indirect effect of forgiveness on flourishing via self-warmth and self-coldness was found to be significant, as the 95% confidence interval did not include zero (B=26. CI95%[.157, .363]).” What does IE stand for in the following sentence? “However only self-warmth was significant mediator (IE, B=.26; CI95%[.189, .344]).What does IE stand for in the following sentence? “However only self-warmth was significant mediator (IE, B=.26; CI95%[.189, .344]).” I believe the manuscript can be accepted after minor revision (corrections to minor methodological errors and text editing).
Answer: I have completed missing information.

Reviewer 2 Report
1. It is recommended that you delete data with questionnaire scores outside three standard deviations.
2. Please describe the correlation calculation method showed in Table 1.
3. Please further test and indicate whether the data are normally distributed. If not, please use the Spearman rank correlation instead of Pearson correlation.
4. Please explain the results that only self-warmth was significant mediator.
5. Please explain the results that self-warmth, but not self-coldness, was a moderator
6. Perhaps you could examine the role of self-warmth in a moderated mediation model.
7. Much work needs to improve the discussion section. For example, explain the current results based on the previous literature or theories, what is contribution of the current study to this filed.
Author Response
2 Reviewer
Thank you very much for your review. We appreciate your suggestions and comments, which will undoubtedly improve the quality of our manuscript.
- It is recommended that you delete data with questionnaire scores outside three standard deviations.
Answer: Thank you for your recommendation. I used the Mahalanobis distance to find outliers.
- Please describe the correlation calculation method showed in Table 1.
- Please further test and indicate whether the data are normally distributed. If not, please use the Spearman rank correlation instead of Pearson correlation.
Answer: I have changed the calculation to Spearman’s rho.
- Please explain the results that only self-warmth was significant mediator.
Answer: I have added new explanation in discussion.
- Please explain the results that self-warmth, but not self-coldness, was a moderator.
Answer: I have added new explanation in discussion.
- Perhaps you could examine the role of self-warmth in a moderated mediation model.
Answer: Thank you for your suggestion. in future research, I am planning a more complex model that will test the indirect role of self-compassion in a more comprehensive way. I see these results as a prelude to a broader project.
- Much work needs to improve the discussion section. For example, explain the current results based on the previous literature or theories, what is contribution of the current study to this filed.
Answer: Thank you for your comment, I have added new information in article.

Reviewer 3 Report
In this paper the authors analyze the relationship between three main aspect of personality: general tendency to be forgiving, self-compassion and flourishing (as the ability for a person to grow as a human being through good times and through life struggles). The sample is big (n=300) and the paper is well written. The result showed that the dimensions are strongly intertwined one over another, in particular positive self-compassion correlate with positive resources (flourishing). This results is in line with the literature and, even if predictable in term of psychological function, this paper demonstrated this correlation with psychometric test.
Author Response
Thank you for your review.
Round 2
Reviewer 2 Report
1. What’s this study extent the previous findings need to specify in the discussion.
2. I don’t think so many subheadings needed in the introduction part.
3. Some framework of this MS needs to improve, for the abstract: the authors should present the aim of this study, what is the scientific question in this study. And the method part, the author should report what kind of data analysis used to explore this question. Not only report the participants and the questionnaires used in this study. In the last par of the abstract, authors need to made a conclude of these results. That means what is the implications of this study. For the conclusion part, the authors should not put the limitation in this part.
Author Response
Thank you very much for your review. We appreciate your suggestions and comments, which will undoubtedly improve the quality of our manuscript.
I have completed the missing data in the abstract and deleted some subtitles in introduction. Also the discussion has been revised to refer to previous studies and to indicate what the current study has contributed to understanding the relationship between the variables under investigation.